# Mega Meta-QTLs: A Strategy for the Production of Golden Barley (*Hordeum vulgare* L.) Tolerant to Abiotic Stresses

**DOI:** 10.3390/genes13112087

**Published:** 2022-11-10

**Authors:** Mahjoubeh Akbari, Hossein Sabouri, Sayed Javad Sajadi, Saeed Yarahmadi, Leila Ahangar, Amin Abedi, Mahnaz Katouzi

**Affiliations:** 1Department of Plant Production, Collage of Agriculture Science and Natural Resource, Gonbad Kavous University, Gonbad-e Kavus 4971799151, Iran; 2Horticulture-Crops Reseaech Department, Golestan Agricultural and Natural Resources Research and Education Center, Agricultural Research, Education and Extension Organization (AREEO), Gorgan 4969186951, Iran; 3Department of Plant Biotechnology, Faculty of Agricultural Sciences, University of Guilan, Rasht 4199613776, Iran; 4Crop Génome Dynamics Group, Agroscope Changins, 1260 Nyon, Switzerland

**Keywords:** barley, abiotic stresses, meta-analysis, QTL, MQTL, consensus map

## Abstract

Abiotic stresses cause a significant decrease in productivity and growth in agricultural products, especially barley. Breeding has been considered to create resistance against abiotic stresses. Pyramiding genes for tolerance to abiotic stresses through selection based on molecular markers connected to Mega MQTLs of abiotic tolerance can be one of the ways to reach Golden Barley. In this study, 1162 original QTLs controlling 116 traits tolerant to abiotic stresses were gathered from previous research and mapped from various populations. A consensus genetic map was made, including AFLP, SSR, RFLP, RAPD, SAP, DArT, EST, CAPS, STS, RGA, IFLP, and SNP markers based on two genetic linkage maps and 26 individual linkage maps. Individual genetic maps were created by integrating individual QTL studies into the pre-consensus map. The consensus map covered a total length of 2124.43 cM with an average distance of 0.25 cM between markers. In this study, 585 QTLs and 191 effective genes related to tolerance to abiotic stresses were identified in MQTLs. The most overlapping QTLs related to tolerance to abiotic stresses were observed in MQTL6.3. Furthermore, three MegaMQTL were identified, which explained more than 30% of the phenotypic variation. MQTLs, candidate genes, and linked molecular markers identified are essential in barley breeding and breeding programs to develop produce cultivars resistant to abiotic stresses.

## 1. Introduction

Increasing biotic and abiotic stresses have a negative impact on the yield and productivity of agricultural products in the world [1,2]. Crops are susceptible to environmental stress, and these changes affect their performance and development. Environmental stresses can be divided into two categories: abiotic stresses, including salinity, drought, temperature, and biotic stresses, including fungi, bacteria, and nematodes [1,3]. Barley is a suitable crop for genetic research due to its high adaptability, low chromosome number, diploid, ease of cross-breeding and cultivation in a wide range of climatic conditions, and tolerance to salinity, drought, and fungal diseases [4,5].

Plants are generally more sensitive to abiotic stresses in the seedling emergence and early seedling growth stages. Moreover, plants are more tolerant to abiotic stresses in the germination stage. The sensitivity of the plant constantly changes during the growing stages. Abiotic stress tolerance contains several quantitative and physiological traits controlled by multiple QTLs [6,7,8].

Improving quantitative traits via plant breeding is difficult. Thus, identifying individual genes with the most effect on phenotype is challenging [9].

The development of molecular techniques such as QTL is a common and powerful tool for explaining complex traits and identifying chromosomal regions containing candidate genes associated with complex traits [10,11]. The advent of the first molecular markers was accompanied by dramatic advances in QTL mapping.

The first genetic map of the *Hordeum vulgare* L. was developed using the RFLP marker. Then, markers such as SNP, DArT, SSR, and AFLP were used to construct linkage maps. Linkage maps are used to identify QTLs and organize the genome, and higher-density maps are used to identify molecular markers associated with QTLs [12].

A meta-analysis is the collection and synthesis of scientific research results and their analysis, and ultimately the presentation of more information than available findings. This technique was first introduced in the 1970s [13,14,15]. Many QTLs were identified in various experiments [16]. QTL meta-analysis integrates QTL data. In this regard, statistical power to identify QTLs increases and improves the accuracy of identifying candidate genes. Estimating genetic structure in meta-analysis is better than in independent research. Reducing confidence intervals for QTLs is another benefit of QTL meta-analysis [17,18].

Ellis et al. identified 12 QTLs for seven traits in barley. They reported QTLs for leaf emergence, stem weight, and the number of tillers for the first time [19]. Five QTLs were identified for salinity tolerance on chromosomes 1H, 2H, 5H, 6H, and 7H, some of which were mapped to other related cereals. These QTLs account for more than 50% of phenotypic variations [20]. Sbei et al. examined salinity-tolerant QTLs using 384 SNP markers, of which seven major QTLs identified effects on chromosomes 1, 2, 3, 4, and 5 [21]. Ghaffari Moghadam et al. identified 23 QTLs related to salinity tolerance at five salinity levels (4, 8, 12, 15, and 20 dS/m). These QTLs explained 9.1 to 15.4% of the phenotypic variance of the traits [22]. Using the double haploid population from the crossing of CM72 (salinity tolerant) and Gairdner (salinity sensitive), 30 QTLs were identified for ten traits that explained 3.25 to 29.81% of the phenotypic variance. QTLs identified under salinity stress on chromosome 4 were associated with spike numbers [23]. Other studies identified QTL for many traits related to salinity abiotic stress [24,25,26,27,28,29,30]. Makhtoum et al. investigated the Iranian barley population using 103 samples from F8 families derived from the cross of two cultivars of Badia×Kavir. They identified 26 QTL-controlling traits under salt stress, 9 QTL-controlling traits under drought stress, and 8 QTL under normal conditions [31,32].

Drought stress is economically the most important abiotic stress that limits production and reduces the quality, nutritional value, and grain yield [33,34]. Using RIL populations, Gudys et al. identified 64 QTL populations for 25 physiological and biochemical traits [17]. In a study of 79 identified QTLs, 55 QTLs for stem traits, 15 QTLs for root traits, and 9 QTLs for physiological traits were detected, most of which are located on chromosomes 1H, 2H, 4H, and 5H [33]. Using 301 BC2DH lines from the cross between Scarlett cultivar and wild barley ISR42-8, Arifuzzaman et al. evaluated six traits in two controlled and drought-stress environments over three years using 371 DNA markers, which led to the identification of 33 QTL [35]. Sayed et al. identified four QTLs related to proline content (PC) on chromosomes 3H, 4H, 5H, and 6H and four QTLs related to leaf wilting (WS) on chromosomes 1H, 2H, 3H, and 4H [36]. In a study, 68 QTLs with a LOD of more than 2.5 were identified, and one to 12 QTLs were detected per trait. The results showed the effect of several genes on different traits in the barley genome [34].

Huang et al. (2018) identified Mn^2+^ controlling QTLs based on chlorophyll content and plant survival using the DH population caused by Yerong and Franklin crosses (waterlogged and susceptible, respectively). The seven QTLs identified on chromosomes were 1H, 3H, 4H, and 6H [37]. In an experiment conducted in 2005 and 2008 using the RIL population-caused Prisma and Apex crosses, 41 QTLs were identified for 18 traits, which explained 8.4 to 54.4% of the variance of the traits [38]. Another study identified 4 QTL on chromosomes 2H, 3H, and 4H [39]. Saal et al. identified 82 QTLs for 1000-seed weight, day to maturity, plant height, number of spikes per square meter, and grain yield using the BC2DHS42 population [40]. Yang et al. QTLs for grain protein concentrations were detected using 146 recombinant inbred lines caused by ‘Lewis’ (CI15856) and ‘Karl’ (CI15487) [41].

Li et al. used two different populations in two seasons. Genetic linkage map made with SSR, RFLP, and DArT markers to identify quantitative trait (QTL) loci associated with traits related to waterlogging tolerance (such as leaf chlorosis, plant survival, and biomass reduction). They found 20 QTLs to waterlogging stress, some of which affected several traits [42].

A study using 156 double haploid lines caused Yerong (waterlogging tolerant) and Franklin (waterlogging sensitive) to identify 31 QTLs, which explained from 4.74 to 55.34% phenotypic variations [43]. Zhou et al. produced a consensus map using six double haploid populations and identified four QTLs that explained 6.2 to 30.1% phenotypic variations [44]. The first report of QTL for root porosity in the barley, which describes the main mechanism of waterlogging tolerance, identified a QTL for root porosity on chromosome 4H, which explains 35.7 and 39% of phenotypic variations in control and hypoxia [45]. Low temperatures can limit the yield of many crops, and a better understanding of genetics can be effective in this challenge [46]. Skinner et al. identified four QTLs under low-temperature stress conditions by preparing a genetic map using populations caused by Dicktoo and Morex crosses [47].

Chlorophyll fluorescence is one of the most useful techniques for obtaining accurate information on the status of the photosystem in cost-effective plant physiology under abiotic stresses [48]. QTLs related to flag leaf length, width, and chlorophyll content detected using a double haploid population caused Yerong and Franklin crosses, and 9 QTLs were identified on five chromosomes that explained 1.9 to 20.2% phenotypic variation of traits [49].

In this study, the analysis of Meta-QTLs abiotic stress tolerance was performed using many identified QTLs. Moreover, candidate genes for each Meta-QTL were identified in the annotated Gene Ontology (GO) database. This study’s results help improve climate modification strategies to resist abiotic stresses. The selection based on the markers connected to the large effect Meta-QTLs known in this study can help us obtain a plant resistant to most abiotic stresses. We named this plant Golden Barley.

## 2. Materials and Methods

### Construction of a Consensus Linkage Map

This meta-analysis included data for salinity, drought, waterlogging, mineral deficiency tolerance, and low-temperature stresses. In order to identify Meta-QTLs affecting abiotic stress in barley (*H. vulgare* L.), first, the necessary data were extracted from several articles (Figure 1).

The abiotic tolerance QTLs in this study were collected from previously published papers (Table 1). We obtained 1162 original QTLs for 116 different traits that were originally mapped from different populations (Table 2). Table 1 shows the traits used in QTL mapping for abiotic stress tolerance in previous studies. In addition, basic information, such as mapping method, flanking markers, most likely position, 95% confidence interval (CI), LOD value, and R^2^, were collected for meta-analysis and overview analysis.

A consensus genetic map containing AFLP, SSR, RFLP, RAPD, SAP, DArT, EST, CAPS, STS, RGA, IFLP, and SNP markers was provided. For this purpose, two genetic linkage maps were used as reference [50,51]. However, the number of markers common between the individual studies used here and any available reference maps was insufficient for a reliable projection of QTL positions. Therefore, BioMercator [52] was used to assemble an integrated map with these published consensus linkage maps [51,53,54] and 26 individual linkage maps (Appendix A).

All calculations for creating the pre-consensus map were performed by applying a weighted least square strategy for marker ordering and determining their position on the consensus map. Individual genetic maps from individual QTL studies were integrated into the pre-consensus map. Chromosomes connected with less than two common markers to the pre-consensus map were excluded before creating the consensus map. Inversions of marker sequences were filtered by discarding inconsistent loci except for very closely linked markers. If the distance of pairwise inversed markers was smaller than 1 cM, only one of the markers was discarded to retain a maximum number of common markers. The consensus map covered a total length of 2,124.43 cM with an average distance between markers of 0.25 cM (Appendix A; Figure 2).

From the 1162 QTL collected, 585 could be projected onto the consensus map generated, given that the markers used to map them were present in the consensus map. For the projection of these QTLs, the confidence interval of 95% was initially calculated for each locus via the equations below: These equations have been modeled for each mapping population [55,56]: F2 and Backcross:(1)$$CI=\left(\frac{530}{number\;of\;lines\times R^{2}}\right)$$
(2)$$\mathrm{or}\;\mathrm{RILs}\;CI=\left(\frac{163}{number\;of\;lines\times R^{2}}\right)$$
(3)$$\mathrm{and}\;\mathrm{for}\;\mathrm{DH}\;CI=\left(\frac{528730}{number\;of\;lines\times R^{2}}\right)$$

Next, the QTL, represented by their middle points and their calculated confidence intervals, original LOD score, and R² were directly projected onto the consensus map. The meta-analysis was carried out, individually by chromosome, through the Veyrieras two-step algorithm in the software [75]. The Akaike (AIC) statistics were used to define the best model for the definition of the number of Meta-QTL or “real” QTL, which best represent the original QTL.

The algorithms and statistical procedures implemented in this software are well-described in the literature [52,75,76]. All files prepared to run BioMercator V.4.2, i.e., maps and QTL files for each barley chromosome, are made available in Appendix A.

## 3. Results

### 3.1. Distribution of QTLs and MQTLs

In total, forty-eight of the 65 QTL analysis studies contained all the information needed for MQTL analysis. A total of 585 QTLs (206 for salinity, 210 for drought, 47 for waterlogging, 106 for toxicity and mineral deficiency, and 16 for low-temperature stresses) were identified from 37 experiments under meta-analysis and projected on the consensus map (Table 3 and Figure 3). The number of QTLs controlling traits related to abiotic stress tolerance ranged from 58 QTLs (on chromosome 5H) to 120 QTLs (on chromosome 2H). The details of the identified meta-QTLs, including the number of initial QTLs, their position in the consensus map, flanked markers, CI and R^2^, are shown in Appendix A.

The number of QTLs in each MQTL for tolerance to abiotic stresses ranged from 1 QTL in MQTL4.4, MQTL5.5, and MQTL5.7 to 38 QTLs in MQTL6.3 (Appendix A).

Most of the QTLs (43 QTLs) for salinity stress were located on chromosome 2. Moreover, the highest QTLs were located on chromosome 4 (9 QTLs), chromosome 1(11 QTLs), and chromosome 3 (25 QTLs) for low temperature, waterlogging, and mineral deficiency stresses, respectively.

Based on our meta-analysis, 128 MQTLs were identified for 585 QTLs related to abiotic stress tolerance. The number of MQTLs per chromosome ranged from 12 MQTLs (on chromosome 6) to 23 MQTLs (on chromosome 2). Moreover, 44, 32, 23, 23, and 6 MQTLs were respectively identified for salinity tolerance, drought tolerance, waterlogging tolerance, mineral deficiency tolerance, and low-temperature tolerance. The number of QTLs and MQTLs for each chromosome is shown in Table 4 and Appendix A.

Li et al. (2013) identified 79 MQTLs for 337 QTLs associated with abiotic stress tolerance traits. The number of these MQTLs varied from 7 (on chromosome 6) to 20 (on chromosome 2). They identified 17 MQTLs for waterlogging stress tolerance, 26 MQTLs for drought stress, 22 MQTLs for salinity, 11 MQTLs for low temperature, and 3 MQTLs for toxicity and mineral deficiency [16]. Zhang et al. (2016) studied 195 QTLs to identify salinity, drought, and waterlogging stress tolerance in barley. Their meta-analysis identified 37 MQTLs for abiotic stress tolerances [53].

### 3.2. Overlap of QTLs in Meta-QTLs

The highest QTL overlap was seen in MQTL6.3 (19 QTL drought tolerant, 9 QTL toxicity and mineral stress elements, 1 QTL waterlogging tolerance, and 9 QTL for salinity stress tolerance). In MQTL4.8, QTLs associated with all abiotic stresses existed (18 QTL drought tolerant, 2 QTL low-temperature tolerance, 3 QTL toxicity and deficiency of mineral element, 3 QTL waterlogging tolerance, and 10 QTL to salinity tolerance). In MQT2.9, MQTL4. 1, MQTL5.1, MQTL5.3, and MQTL7.5, only QTLs associated with salinity tolerance were observed. Moreover, in MQTL7.2, MQTL7.7, and MQTL7.7, the identified QTLs were related to drought tolerance, toxicity, deficiency of mineral elements, and waterlogging stress. Among the identified MQTLs, MQTL4.4, MQTL5.5, and MQTL5.7 contained only one QTL each (Table 5).

Li et al. (2013) tracked 18 MQTLs for abiotic stresses, many of which overlapped. Chromosomes 7H, 2H, 3H, 4H, 1H, and 5H were identified as regions 2, 5, 2, 4, 4, and 1. These regions contained several overlapping MQTLs. Of the 18 MQTLs detected in three regions of the barley genome, an overlap was observed between W-MQTL (water-logging) and T-MQTL (low temperature). Additionally, in the four regions between S-MQTL (salinity) and T-MQTL, in the three regions between W-MQTL and D-MQTL (drought), in the five regions between S-MQTL and D-MQTL, in the four regions between W-MQTL and S-MQTL, in two regions between T-MQTL and D-MQTL, and finally in an overlap region between S-MQTL and M-MQTL (mineral toxicity and deficiency). In the study of Li et al. (2013), MQTL on chromosome 6 was not detected [16]. In the study by Zhang et al. (2016), the highest overlap was reported between QTLs controlling abiotic stress tolerance on chromosome 4 at a genetic distance of 78.61 to 117 cM. There was also an overlap of QTLs on chromosomes 1H and 7H related to salinity, drought, and waterlogging tolerance. Most of the QTLs associated with salinity stress tolerance on chromosome 2H were in the genetic distance of 0 to 53.82 cM [53].

### 3.3. Major MQTLs

In this study, we identified eight MQTLs with R^2^ values of more than 20%. MQTL6.4 explained 26% of the phenotypic changes. This MQTL included two QTLs associated with drought stress, two QTLs associated with toxicity and mineral deficiency, and nine QTLs associated with salinity stress. In MQTL5.8, 11 QTLs for shoot dry weight, root-to-shoot ratio, proline content, root length, shoot dry weight/plant, and harvest index were identified for drought stress. Moreover, one QTL for leaf injury score under salinity stress controlled 26% of phenotypic variation [77].

In total, thirty-two QTLs detected in drought stress, toxicity and mineral deficiency, waterlogging, and salinity were included in MQTL3.2, which explained 23% of phenotypic variation.

In MQTL1.2, 14 QTLs were located that controlled different traits under drought, waterlogging, and salinity stresses and explained 22% of phenotypic variation. Two QTLs were located to control root dry weight under drought stress. Root length, root dry weight, and root-shoot ratio are some of the traits that help the plant to tolerate drought stress [35]. Additionally, the relative water content and stomatal conductance, osmotic regulation, deep root, high shoot weight, high root volume, and high root weight have a positive correlation with stress tolerance [78].

The diversity generated by MQTL1.3 was controlled by several QTLs, some of which controlled drought stress traits. We found that one QTL controlled the traits under toxicity and mineral stress, 3 QTLs controlled traits under water stress, and 10 QTLs controlled traits under salinity stress. MQTL1.3 explained 22% of the phenotypic variation. The QTLs under salinity stress were related to stomata length. Speed and accuracy of measuring stomatal traits are the main obstacles to its use in breeding programs. Stomatal traits are length and width, stomata pores, length, width, and volume of guard cell, length, stomatal density, and index). However, these traits significantly contribute to the barley’s salinity tolerance and grain yield [79].

In total, eighteen QTL controlling traits under drought stress, toxicity and mineral deficiency, flooding, and salinity could explain 22% of MQTL1.6 phenotypic variation.

MQTL2.5 could explain 22% of the phenotypic variation and included 14 QTLs related to drought stress, five QTLs related to waterlogging stress, and 4 QTLs related to salinity stress. Moreover, two, two, and one QTLs controlling grain yield, spike length, and leaf chlorosis were detected under waterlogging conditions. Higher values of plant height, peduncle length, leaf area, ear length, number of seeds, dry weight, grain yield, harvest index, potassium accumulation (K^+^), and the potassium to sodium concentration ratio (K^+^/Na^+^) belong to salinity tolerant genotypes. Therefore, the use of MQTLs containing these traits can be used in breeding programs [80].

In total, twelve QTLs were located in MQTL7.6, which only controlled traits under drought and salinity stresses. This MQTL was able to explain 22% of the phenotypic variation.

### 3.4. Candidate Genes

The total length of the barley genome map used in this study is approximately 21,234.43 cM. In this study, a total of 2593 candidate genes were placed in 52 MQTLs related to abiotic stress tolerance, with an average of 4.86 candidate genes per MQTL. Of the identified candidate genes, 306, 512, 695, 247, 176, 241, and 416 were located on chromosomes 1 to 7, respectively. MQTL3.6, MQTL6.1, and MQTL6.2 had the highest number of genes, with 407, 115, and 109 genes, respectively.

Among the identified candidate genes, 192 genes were significant. MQTL3.6, with 41 genes, contained the most significant candidate genes, followed by MQTL2.1 and MQTL2.2, with 16 and 15 significant candidate genes. In MQTL1.2, MQTL1.3, MQTL3.1, MQTL3.2, MQTL4.1, MQTL4.2, MQTL4.3, MQTL4.4, MQTL4.5, MQTL4.6, MQTL5.2, MQTL5.5, MQTL5 6.6, MQTL5.7, MQTL5.8, MQTL5.9, MQTL6.3, MQTL7.6, MQTL7.7 and MQTL7.8, the identified candidate genes were non-significant (Appendix A).

### 3.5. Evaluation of MQTLs in Geneinvestigator Software

Examination of the geneinvestigator software showed that 191 effective genes for tolerance to salinity abiotic stresses were identified in the MQTLs of this study. Of the 191 candidate genes, 24, 43, 61, 7, 9, 15, and 17 genes were located on chromosomes 1 to 7, respectively. MQTL3.6, MQTL2.2, and MQTL2.1 had the highest number of genes, with 38, 16, and 15 genes, respectively. The expression of genes under abiotic stresses under shoot stress treatment (35.5 h), root stress treatment (35.5 h), PEG stress treatment (6 h), PEG stress treatment (24 h), NaCl stress treatment (6 h), NaCl stress treatment (24 h), PEG + NaCl stress treatment (6 h) and PEG stress treatments were investigated. The highest significant gene expression was observed in MQTL3.6, of which 51 genes had a significant increase in expression and seven genes had a significant decrease in expression. The least significant genes were found in MQTL2.7, MQTL2.8, MQTL5.4, and MQTL6.4. In each of the above MQTLs, only one gene with a significant decrease in expression was observed (Appendix A; Figure 4).

## 4. Discussion

### 4.1. Mega MQTLs

#### 4.1.1. Mega MQTL6.3

Mega MQTL6.3 had the greatest impact on explaining variation by clarifying more than 55% of phenotypic variation. This mega MQTL consists of 19 QTL associated with drought tolerance, 9 QTL with toxicity and mineral deficiency tolerance, 1 QTL with waterlogging tolerance, and 9 QTL with salinity tolerance.

There are essential drought tolerance traits in this region in which QTL played an important role: proline content, thousand kernel weight, grain yield, shoot dry weight, number of tillers, number of kernels, harvest index, number of spikes, and flag leaf width. In fact, drought tolerance is a multigene trait, and identifying the genetic structure helps describe the gene network that controls drought tolerance [53,81]. Proline accumulation is an adaptive metabolic reaction to drought stress. Drought gradually decreases the water content and increases the proline and abscisic acid content in the roots and leaves of genotypes [82]. Leaf wilting score is a simple measure of a plant’s ability to tolerate drought that is associated with drought stress-induced metabolites such as proline. Studies showed a significant correlation between proline content and leaf wilting score [36]. Barley is relatively sensitive to water restriction in the stages of stem formation, grain formation and filling, and with a significant decrease in relative water content, number of fertile spikes, number of seeds per spike, 1000-seed weight, grain, and biomass yield, and increase in ion leakage. The amount of proline is associated with catalase and guaiacol peroxidase enzyme activity in flag leaves [49]. The significant QTLs identified in this MQTL, which are related to proline content, are QPC.S42.6H and QPC.S42.6H.

Numerous QTLs were identified for salinity stress tolerance in this MQTL [78,80]. The important QTL detected in this MQTL is qSHT-6b. Salinity is one of the most important abiotic stresses that affect barley grain yield and quality [83]. Salinity above the threshold level causes osmotic and ionic stress in the plant and significantly reduces plant yield [84]. The most negative effect of salinity stress on the plant occurs during the early stages of growth and germination. Salinity tolerance is a complex physiological trait involving several mechanisms [85].

QTLs have been identified in this MQTL for nitrogen uptake, storage, and remobilization and their relationship to agronomic characteristics. The genetic correlation between traits was different at different nitrogen application levels. In other words, nitrogen consumption positively affected the correlation between traits [40].

Nitrogen is one of the essential elements needed by plants to increase yield. Absorption of this vital element into the plant is done in three stages: uptake, assimilation, and remobilization.

The only QTL related to waterlogging tolerance detected in this MQTL can be referred to QWl.YyFr.6H. Barley (*H. vulgare* L.) is one of the most important crops in the world that is very sensitive to waterlogging stress. Waterlogging causes multiple plant reactions and creates complex non-biological stresses with many factors such as temperature, plant growth stage, nutrition, and soil type. An efficient approach to withstand this stress is to produce tolerant varieties.

The double haploid lines resulting from the Erleng × Franklin intersection show a significant correlation between salinity tolerance and waterlogging tolerance. Genetic evidence suggests that salinity-tolerant and waterlogging-tolerant QTLs share some physiological mechanisms [44].

#### 4.1.2. Mega MQTL4.8

After Mega MQTL6.3, the highest percentage of diversity justification (33%) belonged to Mega MQTL4.8. In this Mega MQTL, 36 QTLs, including 18 QTLs for drought tolerance, 2 QTLs for low-temperature tolerance, 3 QTLs for tolerance to toxicity and mineral deficiency, there were 3 QTLs for waterlogging tolerance, and 10 QTLs for salinity tolerance.

The first part of the plant that perceives drought stress signals is the roots [86]. Abiotic stresses generally affect the roots more than the shoots. Due to the limited access to the roots compared to the shoot and leaf of the plant, significantly less study has been done on the effect of abiotic stresses on root structure and development. The size and structure of the root systems are important agronomic characteristics of a plant. The roots perform many essential functions, including the uptake of water and nutrients, the storage of nutrients, plant resistance in the soil, and the establishment of biological interactions in the rhizosphere. In drought conditions, root traits are positively and significantly correlated with yield.

Root structure development can vary according to the physical characteristics of the soil, such as soil depth, the presence of impermeable layers, and the amount of moisture in the growing environment. Lack of water and nutrients limits grain growth and grain yield in many ecosystems [87]. Among the QTLs located in this MQTL, we can mention QRdw.S42.4H, which is in the confidence range of 116.978 and 145.021. This QTL had genetic control of root length, root shoot ratio, and thousand kernel weights [35]. Important QTLs detected in this MQTL can be referred to as QSPS.S42.4H.a, QSPS.S42.4H.b, and QTILS.S42.4H.a.

The qCOLD-4S in this MegaMQTL has been shown to control low-temperature tolerance stress at confidence levels of 0 and 47.423 [47]. Moreover, two QTLs identified in this MQTL are qCOLD-4S and qCOLD-4L. Low temperature is one of the major biological stresses that affect the geographical distribution of plants and even plant life. Chlorophyll content and photosynthesis significantly decrease under temperature stress [88]. The results showed that not all barley chromosomes played a role in low-temperature tolerance, which was consistent with the Li et al. study [16].

Among the mineral elements, nitrogen has a greater effect on plant height, leaf chlorophyll content, and barley yield. It is also effective in improving the yield of wheat components such as biochemical reactions, photosynthesis intensity, growing period, and accumulating dry matter in the shoots. However, excessive use of thresholds has a negative effect on barley grain quality [89]. The results showed that not all barley chromosomes played a role in low-temperature tolerance, which is consistent with the Li et al. study [16]. Three QTLs detected in this MQTL are QYld.S42.4H.a, QYld.S42.4H.b, and QYld.S42.4H.c.

MQTL4.8 included 2 QTLs (yfy2.1-3, yfy2.2-3) for leaf yellowing ratio and one QTL (yfmas) for reduction of plant biomass in waterlogging stress tolerance [53]. Waterlogging stress causes anaerobic conditions in the roots that occur due to the water saturation of the soil. This stress causes significant damage to the products. Sudden root growth, energy metabolism, aerosol formation, biomass depletion, photosynthesis rate, and plant hormone signaling are different plant responses to this stress. These disorders can lead to reduced yields in barley, wheat, and corn [90]. The lack of oxygen due to waterlogging stress leads to de-nitration and rapid nitrate loss from the soil, reducing the leaf emergence rate. This, in turn, reduces the number of leaves and delays maturation. The plant’s growth, appearance, and physiological function are rapidly affected by nitrate deficiency, and the leaves begin to turn yellow. The level of leaf yellowing is directly related to the duration of waterlogging [91]. Transient reductions in biomass accumulation occur under waterlogging stress, but grain yield depends on plant capacity for post-waterlogging and pre-maturity recovery. If the barley is affected by stress late in its life cycle, it will no longer be able to produce new tillers and compensate for lost stem biomass [92].

In total, ten QTLs were identified in this MegaMQTL to salinity stress tolerance, which controls traits such as seed dry weight, seedling fresh weight, and total leaf number. Moreover, one QTL was identified for the stem diameter trait that explained 10.9% of the phenotypic variation (LOD = 2.515) [7]. Increasing the amount of sodium and potassium ions in the leaves under the influence of salinity stress reduces yield. The decreased growth of the plant’s leaf and shoot also occurs due to receiving a large amount of energy from the aerial parts to counteract the effects of salinity [93]. The qSDW12-4a was identified for the stem diameter trait with a coefficient of determination of 10.6 and a LOD of 2.422, and the parent Badia increased this trait [22].

#### 4.1.3. Mega MQTL3.5

In total, thirty-one QTLs were located in MegaMQTL3.5, which explains 31% of the phenotypic variation. This MegaMQTL includes 5 QTLs to control drought-tolerant traits, 18 QTLs related to stress toxicity and mineral deficiency traits, and 8 QTLs related to salinity stress tolerance.

In drought stress, plants use different strategies or mechanisms for growth and survival, such as growing excess roots to increase water uptake or reducing leaf transpiration from water shortage damage [94]. Strong, positive, and very significant correlations were observed between yield and root length, root dry weight and root–shoot ratio under drought stress [33]. In this MegaMQTL, QTGW.S42.3H was located for the thousand-grain weight trait in this study, which explains 5.85 of the phenotypic variation [33]. Drought stress limits grain filling and reduces thousand-grain weight. Grain yield showed a positive and significant correlation with the performance components the number of grains per ear, grain weight per ear, and thousand-grain weight [95].

Although barley is more sensitive to the toxicity of aluminum among fine-grained crops, tolerance to aluminum varies among different varieties of the barley plant. Preventing root growth and shortening is the first effect of acidic soil toxicity. Additionally, the toxicity of aluminum in acidic soils may limit the absorption of water and nutrients and thus reduce yield [96,97]. In total, two QTLs were located for root length in this MegaMQTL for aluminum tolerance. Aluminum tolerance is controlled by one or more genes and is quantitatively inherited [71]. We located five QTLs in this MegaMQTL, which explains more than 20% of the phenotypic variation. Additionally, eight other QTLs that explain more than 20% of the phenotypic changes were identified in this MegaMQTL to control the traits tolerant to toxicity stress and mineral deficiency. Reducing the number of mineral elements increases the length of the roots and decreases the height of the plant [98].

A QTL was detected in this MegaMQTL to control plant height trait, which explains 14.5% of phenotypic changes. Under salinity stress, plant height had a positive and significant correlation with grain yield, flag leaf width, and grain weight per spike [99]. Due to its genetic and physiological complexity, control of salinity tolerance traits is done by quantitative trait loci (QTLs). Osmotic stress occurs immediately after plant exposure to salinity stress in the root environment. Induction of stomatal closure and its effect on plant cell growth and metabolism, slowing down shoot growth, total leaf area, and biological function [84].

### 4.2. Gene Ontology

Gene ontology (GO) analysis was performed for 52 identified MQTL regions. Gene ontology (GO) analysis was classified into three categories: biological process, molecular function, and cellular components. The most important biological processes involved included the cellular process, metabolic process, response to stimulus, primary metabolic process, organic substance metabolic process, response to stress, catabolic process, and nitrogen compound metabolic process. The main cellular components involved were the cellular anatomical entity, membrane, intracellular anatomical structure, organelle, cytoplasm, and extracellular region. The most important molecular functions are binding, catalytic activity, transferase activity protein binding, organic cyclic compound binding, and heterocyclic compound binding (Appendix A; Figure 5).

Transcription factors (TF), which are responsible for the regulation of metabolism in the stress environment, and genes that encode Late Embryogenesis Abundant (LEA), antioxidant enzymes, osmolytes, and transporters, are the two main categories of proteins involved in stress tolerance [4].

In MQTL3.6, 407 genes were observed, of which 41 genes showed significant expression changes. HORVU.MOREX.r3.3HG0310640 gene in this MQTL explains the transcription factor, TCP. Os01t0924400 gene in rice is similar to this gene, which describes the Similar to Auxin-induced basic helix-loop-helix transcription factor. Wall-associated receptor-like kinases (WAKs) are important candidates for directly linking the extracellular matrix with intracellular compartments and play a role in growth and stress response processes. The WAK gene family has been identified in plants such as *Arabidopsis* and rice [100].

Significant genes related to kinase in barley include HORVU.MOREX.r3.1HG0049480, HORVU.MOREX.r3.3HG0281000, HORVU.MOREX.r3.4HG0379400, HORVU.MOREX.r3.5HG0425730, which are related to the protein kinase domain and wall-associated receptor kinase, galacturonan-binding domain, were identified in this study. Kinase-like receptors are divided into two main groups in terms of their biological role in plants. The first group plays a role in growth and development, and the second group is more active in response to biotic and abiotic stresses [101]. Rajiv et al. (2021) identified 91 WAK (wall-associated kinases) genes in the barley genome, which were classified into five groups and distributed in different chromosomes. The number of WAK genes in rice and barley is very high compared to *Arabidopsis*, suggesting that HvWAKs underwent a duplication event during evolution [102]. Class III peroxidases are secretory enzymes that belong to a ubiquitous multigene family in higher plants and have been identified to play a role in a broad range of physiological and developmental processes [103]. Heterologous expression of peroxidase genes affects the morphology and stress responses of several crops [104]. In total, sixty significant genes related to plant peroxidase were identified in this study for barley, and the genes similar to these genes in rice explain Similar to Class III peroxidase. Cytochrome P450 monooxygenase genes (CYPs) are among the largest gene families in plants, which are effective in various biological processes, including response to biotic and abiotic stresses. Moreover, P450 genes are prone to expanding due to gene tandem duplication during evolution, resulting in generations of novel alleles with neo-function or enhanced function [105].

HORVU.MOREX.r3.6HG0579680 gene in MQTL6.2 explains Cytochrome P450 reaction. In plants, heat shock proteins (Hsps) play an important role in response to various stresses. Hsp20 is the main family of Hsps. Hsp20 is encoded by nuclear genes. Based on predicted subcellular localization, sequence homology, and function, Hsp20s are divided into different subfamilies (CI-CVI, MTI, MTII, ER, CP, and PX). CI-CVI subfamilies are located in cytoplasm/nucleus, MTI and MTII subfamilies are located in mitochondria, ER, CP and PX are located in the endoplasmic reticulum, chloroplast, and peroxisome, respectively. The biological function of Hsp20s in plant protection under various stress conditions has been well documented in several plants, including soybean, tomato, and pepper [106,107,108,109]. In our study, HORVU.MOREX.r3.1HG0037960 gene in MQTL1.4 controls the reaction of α crystallin/Hsp20 domain. Members of the multicopper oxidase (MCO) family of enzymes can be classified based on substrate specificity [110]. The family of multicopper oxidase (MCO) enzymes includes three main groups: laccases, ferroxidases that oxidize ferrous iron, and ascorbate oxidases [111]. In the broadest sense, laccases constitute by far the largest subgroup of MCOs, originating from bacteria, fungi, plants, and insects. Laccase was first discovered in the Japanese lacquer tree *Rhus vernicifera*. MCOs have the ability to combine the oxidation of a substrate with the four-electron reduction of molecular oxygen to water. The electron transfer steps in these redox reactions are coordinated in two copper centers that usually contain four copper atoms. In a redox reaction catalyzed by an MCO, the substrate electrons are accepted at the mononuclear center (type 1 copper atom) and then transferred to the trinuclear cluster (one type 2 and two type 3 copper atoms) as dioxygen. The binding site works by receiving four electrons and reduces molecular oxygen [112]. In this study, the HORVU.MOREX.r3.1HG0049190 gene controlled Multicopper oxidase, type 1.

Sugar transporter proteins (STPs) play essential roles in sugar transport, growth, and development of plants and possess an important potential to enhance plants’ performance of multiple agronomic traits, especially crop yield and stress tolerance. In a comparative genomic study, STP genes were identified in seven representative crops of Gramineae, including *Brachypodium distachyon* (Bd), *H. vulgare* (Hv), *Setaria italica* (Si), *Sorghum bicolor* (Sb), *Zea mays* (Zm). *Oryza rufipogon* (Or) and *Oryza sativa* ssp. japonica (Os) was done, and a total of 177 STP genes were identified and grouped into four clades [113]. In this study, three genes, HORVU.MOREX.r3.2HG0131100, HORVU.MOREX.r3.2HG0131150, and HORVU.MOREX.r3.2HG0131170, explaining the sugar/inositol transporter reaction, were identified in MQTL2.4.

### 4.3. Markers Selection

The selection of superior plants forms the basis of plant breeding, which was based on phenotypic traits for a long time. Selection based on phenotype is effective when the genetic basis of the trait is relatively simple and the genetic effect of the gene is additive. However, many critical agronomic traits are quantitative traits, such as abiotic stress tolerance or traits whose phenotypes are difficult to identify accurately. Therefore, measuring the genetic potential of the trait using phenotype is incorrect, and the selection is inefficient. Genetic breeders use markers to aid selection, genomic selection (GS), genome-wide selection (GWS), marker-assisted complex or convergent crossing (MACC), marker-assisted gene pyramiding (suggest MAGP), and marker-based backcross (MABC). The marker-assisted or marker-based backcross (MABC) is the simplest form of marker-assisted selection. Moreover, marker gene pyramiding (MAGP) is one of the most important applications of DNA markers for plant breeding. This technique has been proposed and applied to increase resistance to diseases and insects by selecting two or more genes simultaneously. Marker-assisted complex or convergent crossing (MACC) can be performed to pyramid multiple genes/QTLs. In this study, we introduced the most important markers to select lines with significant tolerance to all abiotic stresses (Figure 2).

## 5. Conclusions

Developing cultivars resistant to abiotic stresses is one of the critical goals of the agricultural industry worldwide. In this study, the highest QTL was identified for drought tolerance and the lowest for tolerance to low temperatures. The number of these QTLs was different in different chromosomes and MQTLs. Most overlapping QTLs occurred in MQTL6.3. Our findings revealed eight Major MQTLs with R^2^ values of over 20% and three mega MQTLs explaining over 30% phenotypic variation and controlled various traits under abiotic stresses. In mega MQTL4.8, with an R^2^ of 32%, 36 QTLs were observed, which controlled various traits under all abiotic stresses. Moreover, 2593 candidate genes were identified in fifty-two MQTLs for abiotic stress tolerance. Our gene expression analysis indicated that the most significant gene expression was related to MQTL3.6 with 58 genes. The function of the genes identified in this research was investigated with its counterpart in rice.

## Figures and Tables

**Figure 1 genes-13-02087-f001:**
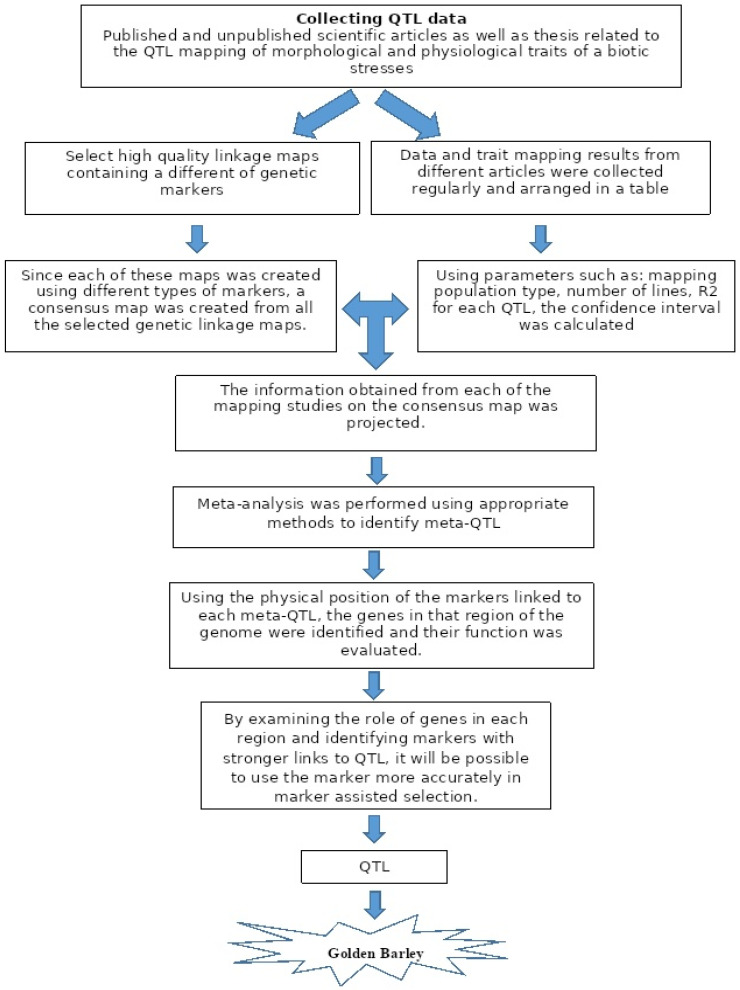
The beginning of a Meta-QTL project to reach the Golden Barley.

**Figure 2 genes-13-02087-f002:**
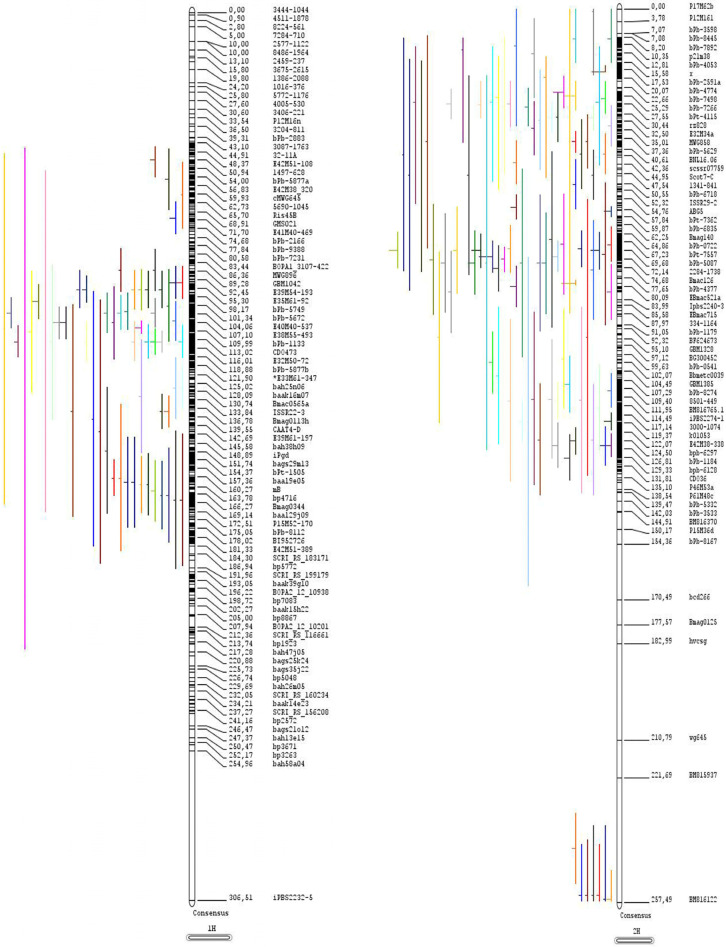
Genomic localization of QTL for abiotic stresses in consensus map.

**Figure 3 genes-13-02087-f003:**
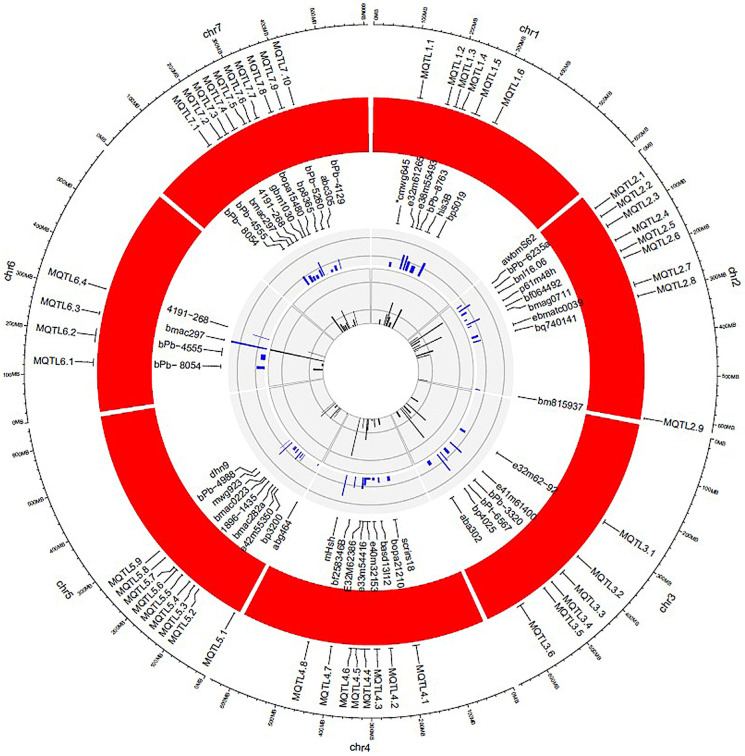
Schematic representation of the distribution pattern of the identified MQTL, QTL, and candidate genes on barley chromosomes. From the center of the plot moving to the outer circle (1) The innermost circle represents the no. of original QTLs (2) R2 of MQTL (3) Marker identified in the major MQTL region (4) MQTLs associated with abiotic stresses (5) Outermost circle represents the barley genome in MB. The red color circle indicates the number of chromosomes in the barley genome.

**Figure 4 genes-13-02087-f004:**
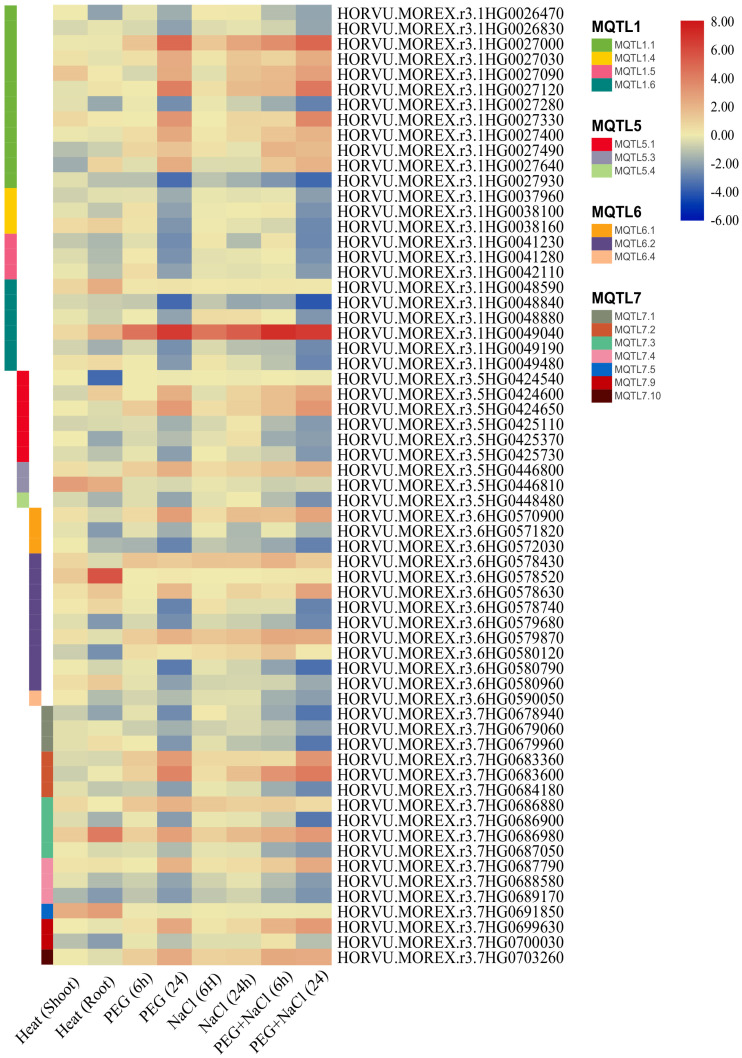
The effective genes for tolerance to abiotic stresses were identified in the MQTLs.

**Figure 5 genes-13-02087-f005:**
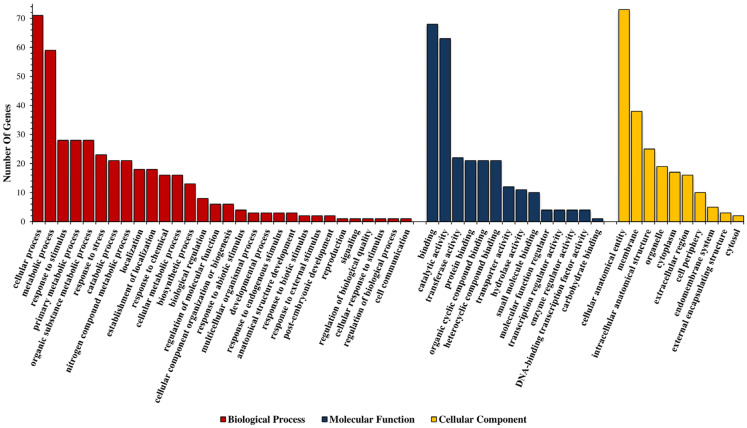
Gene ontology analysis of identified candidate genes in MQTLs.

**Table 1 genes-13-02087-t001:** The abiotic tolerance QTLs in this study were collected from previously published papers.

Reference	Marker	Population	Parents	Population Size	No. of Marker
[6]	SSR, ISSR, iPBS, Scot, IRAP, CAAT	RIL	Badia × Kavir	106	392
[7]	SSR, ISSR, iPBS	F3	Badia × Comino	100	128
[17]	SNP, SSR	RILs	Maresi × Cam/B1/CI08887//CI05761	100	819
[19]	AFLP, SSAP, SSR,	DH	Derkado × B83-12/21/5	156	241
[20]	DArT, SSR	DH	Yuyaoxiangtian Erleng × Franklin	172	858
[22]	SSR, ISSR, iPBS	F3	Badia × Comino	100	128
[23]	SSR, DArT	DH	CM72 × Gairdner	93	332
[24]	SSR, SNP, DArT, STS, CAPS, dCAPS,	DH	Nure × Tremois	118	162
[25]	SSR, DArT	DH	TX9425 × Naso Nijo	188	626
[26]	SSR, DArT, CAPS	DH	Barque-73 × CPI-71284-48	72	1180
[27]	SSR, ISSR, iPBS, Scot, CAAT, IRAP,	RILs	Badia × Kavir	106	302
[28]	SSR	DH	Steptoe × Morex, Harrington × TR306	149 146	103
[29]	DArT, SSR	DH	CM72 × Gairdner	108	886
[30]	DArT, AFLP, SSR	DH	TX9425 ×Franklin	72	520
[33]	SSR, DArT, gene-specific marker	DH	Scarlett × ISR42-8	76	371
[34]	RFLP	RILs	Tadmor × Er/Apm	167	77
[35]	SSR, DArT	DH	Scarlett × ISR42-8	301	371
[36]	SSR, DArT, gene-specific marker	DH	Scarlett × ISR42-8	76	371
[37]	SSR, DArT	DH	Yerong × Franklin TX9425 ×Naso Nijo	177 188	2500 524
[38]	AFLP	RILs	Prisma × Apex R	94	191
[39]	EST, BR, GBM, GBS, RFLP, SSR, SNP	DH	OWBDOM × OWBREC	94	650
[40]	SSR	DH	ISR42-8 × Scarlett	301	98
[41]	SSR	RIL	Lewis(CI15856) × Karl(CI15487)	146	104
[42]	SSR, AFLP, DArT	DH	TX9425 × Franklin, Yerong × Franklin	92 177	520 524
[43]	DArT, AFLP, microsatellite markers	DH	Yerong × Franklin	156	604
[44]	SSR, DArT	DH	YuYaoXiangTian Erleng × Franklin	172	2223
[45]	DArT, SSR	DH	Franklin × YuYaoXiangTian Erleng	126	858
[46]	SSR, RAPD, RFLP, CAPS, AFLP, STS,	DH	Nure × Tremois	136	127
[47]	SSR	DH	Dicktoo × Morex	91	94
[51]	DArT, SSR, RFLP, STS	DH, RIL	Barque73 × CPI71284-48, Clipper × Sahara, Dayton × Zhepi2, Foster × CI4196, Steptoe × Morex, TX9425 × Franklin, Yerong × Franklin	707	2935
[57]	AFLP, SSR, RFLP	DH	Clipper × Sahara 3771	150	211
[58]	RFLP, SSR	F8–9, RIL	Foster × CIho 4196	250	206
[59]	SSR	F4–6, RIL	Fredrickson × Stander	116	143
[60]	RFLP, RAPD, SAP	DH	Steptoe × Morex	150	295
[61]	DArT, AFLP, SSR	DH	TX9425 × Franklin	92	520
[62]	SSR	DH	Steptoe × Morex, Igri × Franka	133	133
[63]	RFLP	DH, F_2_/F_3_	IGRI × FRANKA, VADA × *H. spontaneum*	206	251
[64]	RFLP	DH	PB1 × PB11	111	136
[65]	SSR	DH	Lina × *H. spontaneum* Canada Park	86	325
[66]	SSR, SNP	RILs	ZGMLEL × Schooner	190	1011
[67]	DArT, SNP	RIL	Pompadour × Biosaline-19	98	8610
[68]	RFLP, AFLP, SSR	DH RIL	Steptoe × Morex, Dom × Rec Igri × Franka, L94 ×Vada	317	3258
[69]	SNP, SSR	DH	Huadamai 6 × Huaai 11	122	1962
[70]	SSR	RIL DH	Igri × Franka, Steptoe × Morex, OWBRec × OWBDom, Lina × Canada Park, L94 × Vada, SusPtrit × Vada	645	775
[71]	EST, CAPS, STS, SNP, SSR	DH	Haruna Nijo × H602	93	2948
[72]	RFLP, SSR, AFLP, RGA	DH	Foster × ND9712 × Zhedar	75	214
[73]	RFLP, RAPD, STS, IFLP, SSR, AFLP	DH	Oregon Wolfe Barley	94	830 725
[74]	SNP	RIL	Morex × Barke	81	195

**Table 2 genes-13-02087-t002:** The traits used in QTL mapping for abiotic stress tolerance in previous studies.

Stress	Traits
Drought	Root–shoot ratio, Root dry weight, Plant height, Harvest Index, Leaf weight, Pm.SEVAD, Stomata number, Thousand kernel weight, Leaf wilting score, Leaf osmotic potential, Free proline content, Ethylene content, Light absorption flux (ABS) per PSII reaction center, Water content, Trapped energy flux per PSII reaction center, Root Length, Drought tolerance score, Relative water content, Pm.AUDPCAD, Plant number, No. of Kernels/spike, Number of spikes/plant, Shoot dry weight, Water-soluble carbohydrate concentration at full turgor, Drought water-soluble carbohydrate concentration, Grain yield, Electron transport flux per PSII reaction center (RC), Osmotic adjustment, Osmotic potential under irrigation, Flag leaf weight, Leaf number, Plumule weight, Peduncle length, Flag leaf width, Water loss rate, The average fraction of open RC during the time needed to complete the closure of all RCs, Flag leaf length, Peduncle diameter, Internode length, Water-soluble carbohydrate concentration, REo/RC, DIo/CSm, TRo/RC, ABS/CSo, Fo, TRo/CSo, ABS/CSm, Fm, Sm, DIo/RC, DIo/CSo, ABS/RC
Salinity	GY grain yield, Phenol, Salinity score, Root length, Stomata length, Leaf number, Sugar content, Plumule length, Spike diameter, Peroxidase, Shoot dry weight, Leaf injury score, Stomatal pore area, Spikes per plant, Shoot weight, Biomass, T/C ratio, TR transpiration rate, GS stomatal conductance, Leaf weight, Flag leaf length, Grain weight, Seed dormancy, Catalase, Dry weight per plant, Grain number per plant, Flag leaf width, Plumule weight, Dry weight of roots, Green dry weight of shoots, Green fresh weight of shoots, Fresh weight of roots per plant, Pm.SEVAS, Proline content, Stomata width, Shoot diameter, Tiller number, Seedling weight, Seedling gibberellic acid response, Root dry weight, Leaf length, Flag leaf weight, Na^+^:K^+^ ratio, Awn weight, Seminal roots, Peduncle length, REo/RC, DIo/CSm, TRo/RC, ABS/CSo, Fo, TRo/CSo, ABS/CSm, Fm, Sm, DIo/RC, DIo/CSo, ABS/RC
Waterlogging	Longest root length, Shoot dry weight, Shoot fresh weight, Root fresh weight, Leaf chlorosis, Waterlogging index, Grain yield, Spike length, Plant survival, Tiller number, Leaf yellowing proportion, Porosity, Plant biomass reduction, Grains per spike
Cold stress	Cold score, TMC-Ap3 accumulation, Frost tolerance, Fv/Fm value, plant survival, COR14b accumulation, Vernalization requirement
Nitrogen Deficiency	Thousand-grain weight, Plant survival, Plant height, Number of ears, Grain yield, Days until heading, Thousand kernel weight, Sheath weight, Stem weight, Straw weight, Above-ground nitrogen uptake N, Nitrogen use efficiency of biomass, Above-ground biomass, Leaf weight, CP at maturity carboxypeptidase, Leaf chlorosis, N remobilization efficiency, Grain protein content
Aluminum toxicity	Aluminum tolerance score
Mn toxicity	Leaf chlorosis, Plant survival

**Table 3 genes-13-02087-t003:** R^2^, Meta-QTL, Peak position (bp), Meta-QTL CI range bp, and Closest markers of detected meta-QTLs.

Chr	Meta-QTL	AIC	R^2^ Meta	Meta-QTL Peak Position (bp)	Meta-QTL CI Range bp	Closest Markers
1	MQTL1.1	546.03	0.07	120604292	113763968–128399829	* Cmwg645(59.938)-bPb-0395(60.740)
	MQTL1.2		0.22	191424829	187288939–198521321	E32M61-265(94.97)-SCRI_RS_113745(95.12)
	MQTL1.3		0.22	215895490	209987507–219651209	E38M55-493(107.09)-0501A(107.11)
	MQTL1.4		0.15	227122225	224396043–241295755	bPb-8763(113.65)-E40M32-654(113.84)
	MQTL1.5		0.11	274117654	257758192–281940383	His3B(135.24)-ABC152F(135.81)
	MQTL1.6		0.22	328366571	317252081–331517042	bp5019(162.71)-bPt-5002(163.17)
2	MQTL2.1	966.53	0.09	42367561	39815442–44919679	AWBMS62(17.55)-WG516(17.79)
	MQTL2.2		0.2	68439906	64629701–72250111	bPb-6235a(28.42)-bPb-6128(28.76)
	MQTL2.3		0.09	97387878	91468880–103306875	BNL16.06(40.61)-bPb-3574(41.08)
	MQTL2.4		0.04	139419949	135909288–142930609	P61M48h(58.13)-bPb-3575(59.69)
	MQTL2.5		0.22	170213114	166570654–173855574	BF064492(71.03)-AWBMA33(71.19)
	MQTL2.6		0.06	189767373	187766416–191768330	Bmag0711(49.14)-EBmac521a(80.08)
	MQTL2.7		0.1	263383407	256445958–270320855	EBmatc0039(109.90)-AWBMA21(109.95)
	MQTL2.8		0.12	294464135	290701857–298226412	BQ740141(122.82)-3179-497(122.96)
	MQTL2.9		0.07	611525900	608806036–614245763	BM815937(221.69)-BM816122(257.49)
3	MQTL3.1	775.59	0.09	268408330	261707356–275109306	E32M62-92(101.28)-E35M48-250(101.83)
	MQTL3.2		0.23	379792248	376613242–384250770	E41M61-400(143.95)-E42M51-442(143.96)
	MQTL3.3		0.1	436460332	428189641–444731023	bPb-3320(165.22)-basd27g02(165.47)
	MQTL3.4		0.17	473526748	471178768–475874727	bPt-6567(179.18)-bPt-5150(179.52)
	MQTL3.5		0.31	498826884	495476397–502177372	bp4025(189.03)-7241-553(189.17)
	MQTL3.6		0.07	573909459	563449078–584369838	ABA302(217.25)-P15M47-91(217.59)
4	MQTL4.1	889.8	0.07	182695125	175965050–193845575	SCRI_RS_180891(80.50)-SCRI_RS_119628(83.45)
	MQTL4.2		0.08	250437750	245089075–255786425	BOPA2_12_10063(113.09)-E36M62-78(113.51)
	MQTL4.3		0.04	285624125	277921575–293326675	basd13l12(128.85)-E33M60--5.5(129.43)
	MQTL4.4		0.03	317672025	304090350–331253725	E40M32-153(143.70)-E36M59-94(143.91)
	MQTL4.5		0.12	339354100	332668250–346039950	E33M54-416(153.49)-bPb-6872(153.58)
	MQTL4.6		0.17	352946825	348117550–357776125	E32M62-386(158.61)-E38M55-139(158.80)
	MQTL4.7		0.17	403538300	400996575–406080025	BF258346B(182.46)-bPb-7395(182.61)
	MQTL4.8		0.32	462329625	460130475–465799650	mHsh(209.15)-ABC305(209.24)
5	MQTL5.1	710.31	0.03	11389965	9196970–13582960	ABG464(10.90)-Act8A(11.32)
	MQTL5.2		0.12	121232310	120331310–122133305	bp3200(106.34)-E40M40-354(107.08)
	MQTL5.3		0.09	135954265	133744270–138164260	E42M55-350(119.80)-E37M50-70(120.07)
	MQTL5.4		0.14	152138215	149316225–154960210	Bmac282a(134.20)-E37M62-231(134.43)
	MQTL5.5		0.07	177842140	174833150–180851130	1896-1435(156.80)-bags4p07(157.03)
	MQTL5.6		0.06	197188085	192853095–201523070	Bmac0223(173.52)-CDO57B(174.12)
	MQTL5.7		0.06	206322725	204237395–208408050	MWG923(181.91)-GBM1227(182.46)
	MQTL5.8		0.26	239937290	238061630–241812955	bPb-4988(211.63)-7523-440(211.80)
	MQTL5.9		0.19	262479225	262054225–262904225	dhn9(231.4)-bPb-6367(231.8)
6	MQTL6.1	438.51	0.12	129309961	120005985–138613926	bPb- 8054(38.32)-bPb-9768(38.80)
	MQTL6.2		0.07	198777337	181047449–216507225	bPb-4555(58.80)-GBM1049(58.86)
	MQTL6.3		0.55	256188394	251612394–260764394	Bmac297(75.84)-bPb-1116(75.92)
	MQTL6.4		0.26	320961575	319965327–321957823	4191-268(95.03)-E45M48e(93.05)
7	MQTL7.1	640.38	0.15	172041612	165726420–178356810	bPb- 8054(38.32)-bPb-9768(38.80)
	MQTL7.2		0.11	205841892	197894802–213788982	bPb-4555(58.80)-GBM1049(58.86)
	MQTL7.3		0.07	224776302	218058726–231493884	Bmac297(75.84)-bPb-1116(75.92)
	MQTL7.4		0.06	248606394	240290454–256922334	4191-268(95.03)-E45M48e(93.05)
	MQTL7.5		0.11	278695794	273308316–284083272	GBM1030(124.47)-bPb-1039(124.83)
	MQTL7.6		0.21	293069850	289839600–296300106	BOPA1_5480-826(131.10)-BOPA1_ABC11989-1-2-148(131.14)
	MQTL7.7		0.01	311177142	306538548–319839582	bp8365(138.93)-SCRI_RS_122512(139.52)
	MQTL7.8		0.04	352488594	345402156–359575026	bPb-5260(157.53)-Bmag0174b(157.88)
	MQTL7.9		0.08	381929706	374642082–389217330	ABC305(170.76)-bPb-9269(171.07)
	MQTL7.10		0.16	413807478	411996750-415618206	bPb-4129(185.06)-bp5141(185.26)

**Table 4 genes-13-02087-t004:** Numbers of projected QTL and detected MQTL (in brackets).

Chr	Drought	Low Temperature	Water-Logging	Salinity	Mineral Toxicity and Deficiency	Total
1H	18(4)	2(2)	11(4)	21(6)	12(4)	64(20)
2H	54(6)	1(1)	10(4)	43(8)	12(4)	120(23)
3H	43(6)	0	8(4)	23(5)	25(2)	99(17)
4H	32(4)	2(1)	8(3)	39(8)	15(4)	96(20)
5H	16(4)	9(1)	3(2)	23(6)	7(2)	58(15)
6H	21(2)	2(1)	2(2)	22(4)	12(3)	59(12)
7H	26(6)	0	5(4)	35(7)	23(4)	89(21)
Total	210(32)	16(6)	47(23)	206(44)	106(23)	585(128)

**Table 5 genes-13-02087-t005:** Distribution of abiotic tolerance QTLs per Meta-QTLs.

Meta-QTL.	Stress					Meta-QTL	Stress				
	Low Temperature	Mineral Toxicity and Deficiency	Waterlogging Tolerance	Salinity	Drought		Low Temperature	Mineral Toxicity and Deficiency	Waterlogging Tolerance	Salinity	Drought
MQTL1.1	1	3	2	1	0	MQTL4.6	0	0	0	15	3
MQTL1.2	0	0	5	5	4	MQTL4.7	0	9	4	1	0
MQTL1.3	0	1	3	10	4	MQTL4.8	2	3	3	10	18
MQTL1.4	0	0	0	1	2	MQTL5.1	0	0	0	2	0
MQTL1.5	1	1	0	2	0	MQTL5.2	9	0	0	8	0
MQTL1.6	0	7	1	2	8	MQTL5.3	0	0	0	5	0
MQTL2.1	1	0	0	14	0	MQTL5.4	0	5	0	2	2
MQTL2.2	0	7	0	7	18	MQTL5.5	0	0	0	0	1
MQTL2.3	0	1	0	0	8	MQTL5.6	0	0	2	0	0
MQTL2.4	0	2	2	1	1	MQTL5.7	0	0	1	0	0
MQTL2.5	0	0	5	4	14	MQTL5.8	0	0	0	1	11
MQTL2.6	0	2	0	2	0	MQTL5.9	0	2	0	5	2
MQTL2.7	0	0	2	5	12	MQTL6.1	2	0	1	3	0
MQTL2.8	0	0	1	4	1	MQTL6.2	0	1	0	1	0
MQTL2.9	0	0	0	6	0	MQTL6.3	0	9	1	9	19
MQTL3.1	0	0	1	2	5	MQTL6.4	0	2	0	9	2
MQTL3.2	0	7	4	8	5	MQTL7.1	0	11	2	10	2
MQTL3.3	0	0	2	3	8	MQTL7.2	0	0	0	0	6
MQTL3.4	0	0	1	0	14	MQTL7.3	0	0	1	2	0
MQTL3.5	0	18	0	8	5	MQTL7.4	0	7	1	0	0
MQTL3.6	0	0	0	2	6	MQTL7.5	0	0	0	7	0
MQTL4.1	0	0	0	5	0	MQTL7.6	0	0	0	6	6
MQTL4.2	0	0	1	1	6	MQTL7.7	0	3	0	0	0
MQTL4.3	0	2	0	1	0	MQTL7.8	0	0	0	1	2
MQTL4.4	0	0	0	1	0	MQTL7.9	0	0	1	5	1
MQTL4.5	0	1	0	5	5	MQTL7.10	0	2	0	4	9

## Data Availability

The data presented in this study are available upon request from the corresponding author.

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
