# Peer review of "Mega Meta-QTLs: A Strategy for the Production of Golden Barley (Hordeum vulgare L.) Tolerant to Abiotic Stresses"

_genes, 2022, doi:10.3390/genes13112087_

Round 1
Reviewer 1 Report
Dear authors,
the work “ Mega MQTLs: A strategy for the production of golden barley (Hordeum vulgare L.) tolerant to abiotic stresses” seems to have been carried out competently and properly. Overall, the manuscript is well written and presented.
But some editorial corrections are needed and there are some inconsistencies in the text.
In the Introduction the abbreviation MQTL should be disclosed (meta-QTL).
Golden Barley (as in the Abstract) or golden barley (as in the title)? My opinion that it is better to use “Golden Barley” with quotes everywhere in the MS.
In the text there are spellings meta-QTL (f.e. line 142, 166, 213 ) and Meta-QTL (Table 3 ). If the authors put any meaning into this distinction, it needs to be clarified. If not, it should be avoided. Besides QTL in lower cases there is on the Figure 1 (“markers with stronger links to qtl”)
Also it is not clear why the same traits are written in the text and tables both with lowercase and with uppercase letters, f.e.: line 289 – root length, line 297 and Table 2 – Root Length.
It is unclear why in the Introduction in some cases of citations the authors specify the name of the first author (Line 94 -Kornelia Gudys et al. Identified 64 QTL populations, Line 100 Mohammed A Sayed et al. identified four QTLs), and in other cases, they do not (Line 75 Ellis et al. identified 12 QTLs for seven traits in barley, Line 79 Sbei et al. examined salinity tolerant QTLs.
It looks like carelessness.
Line 220 “37 of the 47 QTL analysis studies contained all the information needed for MQTL 220 analysis.”
But Table 1 lists 49 studies from which the data are taken. It needs to be written more clearly.
Besides it should be “Thirty-seven of the 47 QTL…”
Line 336 Geneinvestigator database
What kind of base is this? What is its e-mail address?
Isn't it a genevestigator.com?
Line 402 In this Mega MQTL, 36 QTL including 19 QTL for drought tolerance, 2 QTL for low temperature tolerance, 3 QTL for tolerance Toxicity and mineral deficiency, there were 3 QTLs for waterlogging tolerance and 10 QTLs for salinity tolerance
19+2+3+3+10= 37 (and in the Table there are 18 QTL for drought tolerance)
There are some typos:
Line 53 Also, Plants are more tolerant in the germination stage should be plants
Line 179 Therefore, biomercator [52] was used to assemble… change to BioMercator
Table 2
Suger content – Sugar?
Peroxidas – Peroxidase?
Seedline gibberellic acid response – seedling?
Leaf weight and Plant leaf weight - are these different traits?
Line 228 Closest markers of deteced should be detected
Line 287 This MQTL included 2, 2, and 9 QTLs associated with drought stress, toxicity, mineral deficiency, and salinity stress, respectively.
Four factors are listed, and only 3 numbers of QTL.
Line 318 belong to salinity tolerant genotypes The dot is absent.
Lines 519, 529 – Arabidopsis should be in italics
Line 556 Rhus vernicifera should be in italics
Some sentences need correction in languge, for example:
It is considered correct to begin a sentence only with a capital number
Lines 220, 293, 296, 305, 311, 315, 321 and others.
Table 1: Select high quality linkage maps containing a different of genetic markers
Line 232 Most QTLs for Salinity stress (was located on chromosome 243 QTLs). ??
Line 254 In MQTL4.8, QTLs associated with all abiotic stresses (18 QTL drought tolerant, 2 QTL low tolerance, 3 QTL toxicity and deficiency of mineral element, QTL water lodging tolerance and 10 QTL to salinity tolerance) were existed.
Line 256 In MQT2.9, MQTL4. 256 1, MQTL5.1, MQTL5.3 and MQTL7.5 only QTLs associated with salinity tolerance observed. Should be were observed.
Line 260 Among the identified MQTLs, MQTL4.4, MQTL5.5 and MQTL5.7 contained only one QTL.
Should be only one QTL each.
Line 305 QTLs under salinity 305 stress was were related to stomata length.
Line 306 Speed and accuracy of measuring stomatal traits (stomata length and width, stomata width/length, stomata pores, length, width and volume of guard cell, length, stomata density and index) are the main obstacles to its use in breeding programs are considered, but these traits significantly contribute to salinity tolerance and grain yield in the barley.
Please try to rephrase this sentence and correct the English.
Line 387 Absorption of this important element in the plant change to into the plant
Line 390 The one QTL related to waterlogging tolerance change to The only QTL
Illustrations
Line 167 “The abiotic tolerance QTLs in this study were collected from previously published papers (Table 3).” It is not clear why first mentioned Table has number 3, and why Figure 3 precedes Figure 2 in the text. It would be better if the numbering was sequential.
Fig. 2. The numbers and letters are blurry. Figure should be enlarged, sharpened and enhanced
“The color density indicates the number of chromosome in the barley genome.” It is not clear which color is meant by.
List of references needs especially careful editing.
Ref. 4 is repeated as ref. 100.
Some references can not be found via Internet (ref 7, 22, 78, 84, 99). Ref. 99 - such an article is not in this journal and the numbering of issues and articles in it is different. If these papers are published in local journals and in Iranian, this should be indicated.
Many double surnames have dots instead of hyphens (refs. 1, 5, 6, 7, 9, 10, 11 and many others)
Italics are absent in some references (Lines 734, 857, 787 ), and vice versa:
Line 44 and 680 (Hordeum vulgare ssp. spontaneum) should be (Hordeum vulgare ssp. spontaneum).
There are some other typos.
Line 620 stresvvcses in plants should be stresses in plants
Line 877 TaCYP81D5, one member in a wheat cytochrome P450gene cluster, confers salinity tolerance via reactive oxygenspecies scavenging change to
TaCYP81D5, one member in a wheat cytochrome P450 gene cluster, confers salinity tolerance via reactive oxygen species scavenging
Author Response
Reviewer 1
The article is interesting and deals with the subject of the production of barley resistant to abiotic stress, which is important due to the observed climate changes. The work is written in a logical and orderly manner. The conducted research is described in a detailed and systematic manner. However, I suggest paying attention to two more things:
- In point 4.1.1. the authors report that there is a significant genetic correlation between the nitrogen content and all the agronomic traits studied. I should write more about it.
Answer:
Corrected
- I also suggest rereading the work in order to eliminate minor linguistic errors.
Answer:
In order to solve the problems of English writing of the text, the manuscript was sent to the accredited institution and the article was completely revised. Its certificate has also been sent
Reviewer 2
Dear authors,
the work “ Mega MQTLs: A strategy for the production of golden barley (Hordeum vulgare L.) tolerant to abiotic stresses” seems to have been carried out competently and properly. Overall, the manuscript is well written and presented.
But some editorial corrections are needed and there are some inconsistencies in the text.
Answer:
We have done our best to correct text inconsistencies.
In the Introduction the abbreviation MQTL should be disclosed (meta-QTL).
Answer:
The abbreviation of MQTL should be disclosed to meta-QTL.
Golden Barley (as in the Abstract) or golden barley (as in the title)? My opinion that it is better to use “Golden Barley” with quotes everywhere in the MS.
Answer:
Corrected throughout the text of the manuscript
In the text there are spellings meta-QTL (f.e. line 142, 166, 213 ) and Meta-QTL (Table 3 ). If the authors put any meaning into this distinction, it needs to be clarified. If not, it should be avoided. Besides QTL in lower cases there is on the Figure 1 (“markers with stronger links to qtl”)
Answer:
There was no representation between meta-QTL and Meta-QTL, so we wrote them in the same way.
QTL in lower cases on the Figure 1, corrected.
Also it is not clear why the same traits are written in the text and tables both with lowercase and with uppercase letters, f.e.: line 289 – root length, line 297 and Table 2 – Root Length.
Answer:
Throughout the article, the names of the attributes were written in the same way, and the first letter of the word was capitalized and the rest were written in lowercase
It is unclear why in the Introduction in some cases of citations the authors specify the name of the first author (Line 94 -Kornelia Gudys et al. Identified 64 QTL populations, Line 100 Mohammed A Sayed et al. identified four QTLs), and in other cases, they do not (Line 75 Ellis et al. identified 12 QTLs for seven traits in barley, Line 79 Sbei et al. examined salinity tolerant QTLs.
It looks like carelessness.
Answer:
The name of first authors removed. But in the second case, I did not understand the referee's question. Both articles are accessible using doi. They are mentioned in the references.
Ellis, R.P.; Forster, B.P.; Gordon, D.C.; Handley, L.L.; Keith, R.P.; Lawrence, P.; Meyer, R.; Powell, W.; Robinson, D.; Scrimgeour, C.M.; Young, G.; Thomas, W.T.B. Phenotype/genotype associations for yield and salt tolerance in a barley mapping population segregating for two dwarfing genes. J. Exp. Bot. 2002, 53, 1163-1176. https://doi.org/10.1093/jexbot/53.371.1163
Sbei, H.; Sato, K.; Shehzad, T.; Harrabi, M.; Okuno, K. Detection of QTLs for salt tolerance in Asian barley (Hordeum vulgare L.) by association analysis with SNP markers. Breed. Sci. 2014, 64, 378-388. https://doi.org/10.1270/jsbbs.64.378
Line 220 “37 of the 47 QTL analysis studies contained all the information needed for MQTL 220 analysis.”
But Table 1 lists 49 studies from which the data are taken. It needs to be written more clearly.
Besides it should be “Thirty-seven of the 47 QTL…”
Answer:
It was a mistake in counting the articles that corrected
Line 336 Geneinvestigator database
What kind of base is this? What is its e-mail address?
Isn't it a genevestigator.com?
Answer:
Geneinvestigator database changed to Geneinvestigator software
GENEVESTIGATOR is a bioinformatics software (https://genevestigator.com/) that enables analysis of deeply curated bulk tissue and single-cell transcriptomic data from public repositories with user-friendly visualization tools. The unique collection of high quality data is queried by researchers for various applications in biomarker and target discovery, diagnostics and in silico modeling. The consistent processing and use of common vocabularies to describe biological experiments allows the user to save time and resources, and to focus on addressing exciting questions spanning different scientific fields from plant biotech to pharma.
Line 402 In this Mega MQTL, 36 QTL including 19 QTL for drought tolerance, 2 QTL for low temperature tolerance, 3 QTL for tolerance Toxicity and mineral deficiency, there were 3 QTLs for waterlogging tolerance and 10 QTLs for salinity tolerance
19+2+3+3+10= 37 (and in the Table there are 18 QTL for drought tolerance)
Answer:
It was a mistake in counting the articles that corrected
There are some typos:
Line 53 Also, Plants are more tolerant in the germination stage should be plants
Answer:
It was a mistake in typing that corrected
Line 179 Therefore, biomercator [52] was used to assemble… change to BioMercator
Answer:
It was a mistake in typing that corrected
Table 2
Suger content – Sugar?
Answer:
It was a mistake in typing that corrected
Peroxidas – Peroxidase?
Answer:
It was a mistake in typing that corrected
Seedline gibberellic acid response – seedling?
Answer:
It was a mistake in typing that corrected
Leaf weight and Plant leaf weight - are these different traits?
Answer:
It was a mistake in typing that corrected
Line 228 Closest markers of deteced should be detected
Answer:
It was a mistake in typing that corrected
Line 287 This MQTL included 2, 2, and 9 QTLs associated with drought stress, toxicity, mineral deficiency, and salinity stress, respectively. Four factors are listed, and only 3 numbers of QTL.
Answer:
This MQTL included 2 QTLs associated with drought stress, 2 QTLs associated with toxicity and mineral deficiency, and 9 QTLs associated with salinity stress.
Line 318 belong to salinity tolerant genotypes The dot is absent.
Answer:
Corrected
Lines 519, 529 – Arabidopsis should be in italics
Answer:
corrected
Line 556 Rhus vernicifera should be in italics
Answer:
corrected
Some sentences need correction in languge, for example:
It is considered correct to begin a sentence only with a capital number
Lines 220, 293, 296, 305, 311, 315, 321 and others.
Answer:
corrected
Table 1: Select high quality linkage maps containing a different of genetic markers
Answer:
The articles listed in this table all have high quality maps. On the other hand, we have mentioned all of them so that other researchers can prepare a consensus map later in their own studies, so we thought that the purpose of some of them would reduce the useful information of the article and we mentioned them all.
Line 232 Most QTLs for Salinity stress (was located on chromosome 243 QTLs). ??
Answer:
Corrected as:
Most QTLs for Salinity stress (43 QTLs) was located on chromosome 2
Line 254 In MQTL4.8, QTLs associated with all abiotic stresses (18 QTL drought tolerant, 2 QTL low tolerance, 3 QTL toxicity and deficiency of mineral element, QTL water lodging tolerance and 10 QTL to salinity tolerance) were existed.
Answer:
In MQTL4.8, QTLs associated with all abiotic stresses (18 QTL drought tolerant, 2 QTL low temperature tolerance, 3 QTL toxicity and deficiency of mineral element, 3 QTL waterlogging tolerance and 10 QTL to salinity tolerance) were existed.
Line 256 In MQT2.9, MQTL4. 256 1, MQTL5.1, MQTL5.3 and MQTL7.5 only QTLs associated with salinity tolerance observed. Should be were observed.
Answer:
corrected
Line 260 Among the identified MQTLs, MQTL4.4, MQTL5.5 and MQTL5.7 contained only one QTL.
Should be only one QTL each.
Answer:
Corrected
Line 305 QTLs under salinity 305 stress was were related to stomata length.
Answer:
Corrected
Line 306 Speed and accuracy of measuring stomatal traits (stomata length and width, stomata width/length, stomata pores, length, width and volume of guard cell, length, stomata density and index) are the main obstacles to its use in breeding programs are considered, but these traits significantly contribute to salinity tolerance and grain yield in the barley.
Answer:
Corrected as:
Two QTLs under salinity stress were related to Stomata length. Speed ​​and accuracy of measuring stomatal traits (Stomata length and width, Stomata width/length, Stomata pores, Length, Width and volume of guard cell, Length, Stomatal density and index) are the main obstacles in plant breeding programs. These traits significantly contribute to salinity tolerance and grain yield in the barley [79].
Line 387 Absorption of this important element in the plant change to into the plant
Answer:
Corrected
Line 390 The one QTL related to waterlogging tolerance change to The only QTL
Answer:
Corrected
Illustrations
Line 167 “The abiotic tolerance QTLs in this study were collected from previously published papers (Table 3).” It is not clear why first mentioned Table has number 3, and why Figure 3 precedes Figure 2 in the text. It would be better if the numbering was sequential.
Answer:
Corrected
Fig. 2. The numbers and letters are blurry. Figure should be enlarged, sharpened and enhanced
“The color density indicates the number of chromosome in the barley genome.” It is not clear which color is meant by.
Answer:
Corrected
Figure 2 is the best output from the software, I tried to make it bigger for better clarity.
List of references needs especially careful editing.
Ref. 4 is repeated as ref. 100.
Corrected
Some references can not be found via Internet (ref 7, 22, 78, 84, 99). Ref. 99 - such an article is not in this journal and the numbering of issues and articles in it is different. If these papers are published in local journals and in Iranian, this should be indicated.
Answer:
All sources were corrected and now they are all traceable on the Internet
Many double surnames have dots instead of hyphens (refs. 1, 5, 6, 7, 9, 10, 11 and many others)
Answer:
Corrected
Italics are absent in some references (Lines 734, 857, 787 ), and vice versa:
Line 44 and 680 (Hordeum vulgare ssp. spontaneum) should be (Hordeum vulgare ssp. spontaneum).
Answer:
Corrected
There are some other typos.
Line 620 stresvvcses in plants should be stresses in plants
Answer:
Corrected
Line 877 TaCYP81D5, one member in a wheat cytochrome P450gene cluster, confers salinity tolerance via reactive oxygenspecies scavenging change to
TaCYP81D5, one member in a wheat cytochrome P450 gene cluster, confers salinity tolerance via reactive oxygen species scavenging
Answer:
Corrected

Reviewer 2 Report
The article is interesting and deals with the subject of the production of barley resistant to abiotic stress, which is important due to the observed climate changes. The work is written in a logical and orderly manner. The conducted research is described in a detailed and systematic manner. However, I suggest paying attention to two more things:
1. In point 4.1.1. the authors report that there is a significant genetic correlation between the nitrogen content and all the agronomic traits studied. I should write more about it.
2. I also suggest rereading the work in order to eliminate minor linguistic errors.
Author Response

(The authors gave the same response as above.)
